# An Anti-Inflammatory Poly(PhosphorHydrazone) Dendrimer Capped with AzaBisPhosphonate Groups to Treat Psoriasis

**DOI:** 10.3390/biom10060949

**Published:** 2020-06-23

**Authors:** Ranime Jebbawi, Abdelouahd Oukhrib, Emily Clement, Muriel Blanzat, Cédric-Olivier Turrin, Anne-Marie Caminade, Eric Lacoste, Séverine Fruchon, Rémy Poupot

**Affiliations:** 1INSERM, U1043, CNRS, U5282, Université de Toulouse, UPS, Centre de Physiopathologie de Toulouse-Purpan, F-31300 Toulouse, France; ranime.jebbawi@inserm.fr (R.J.); emily.clement@inserm.fr (E.C.); severine.fruchon@inserm.fr (S.F.); 2CNRS, UMR 5623, Université de Toulouse, UPS, Laboratoire des Interactions Moléculaires et Réactivité Chimique et Photochimique, IMRCP, 118 route de Narbonne, CEDEX 9, F-31062 Toulouse, France; blanzat@chimie.ups-tlse.fr; 3CNRS, UPR 8241, Laboratoire de Chimie de Coordination, 205 route de Narbonne, BP 44099, CEDEX 4, F-31077 Toulouse, France; abdelouahd.oukhrib@lcc-toulouse.fr (A.O.); cedric-olivier.turrin@lcc-toulouse.fr (C.-O.T.); anne-marie.caminade@lcc-toulouse.fr (A.-M.C.); 4LCC-CNRS, Université de Toulouse, CNRS, 31400 Toulouse, France; 5Imavita S.A.S., Canal Biotech 1, 3 rue des Satellites, Parc Technologique du Canal, F-31400 Toulouse, France; eric.lacoste@imavita.com

**Keywords:** dendrimer, drug candidate, imiquimod mouse model, inflammation, phosphonate, preclinical efficacy, psoriasis, topical application

## Abstract

Dendrimers are nanosized, arborescent macromolecules synthesized in a stepwise fashion with attractive degrees of functionality and structure definition. This is one of the reasons why they are widely used for biomedical applications. Previously, we have shown that a poly(phosphorhydrazone) (PPH) dendrimer capped with anionic azabisphosphonate groups (so-called ABP dendrimer) has immuno-modulatory and anti-inflammatory properties towards human immune cells in vitro. Thereafter, we have shown that the ABP dendrimer has a promising therapeutic efficacy to treat models of acute and chronic inflammatory disorders in animal models. In these models, the active pharmaceutical ingredient was administered systematically (intravenous and oral administrations), but also loco-regionally in the vitreous tissue. Herein, we assessed the therapeutic efficacy of the ABP dendrimer in the preclinical mouse model of psoriasis induced by imiquimod. The ABP dendrimer was administered in phosphate-buffered saline solution via either systemic injection or topical application. We show that the topical application enabled the control of both the clinical and histopathological scores, and the control of the infiltration of macrophages in the skin of treated mice.

## 1. Introduction

The World Health Organization (WHO) describes psoriasis as “a chronic, non-communicable, painful, disfiguring, and disabling disease for which there is no cure, and with great negative impact on patients’ quality of life” [1]. Its prevalence in developed countries is around 3%, for which more than 100 million people are affected worldwide. After decades of basic and clinical research, the physio-pathological mechanisms of this skin disorder remain controversial [2,3]. Alternatively, genetic, environmental, and auto-immune factors have been proposed as starting points of psoriasis. It is likely that all of these factors aggregate at the onset, and then nurture the development and the maintenance of the disease. Despite the fact that psoriasis is not considered a true auto-immune disease [4], clinical and pathological correlations have shown similarities with other auto-immune diseases. Indeed, the implication of pro-inflammatory T cells and innate immune cells such as monocytes and dendritic cells infiltrating the dermis, as well as pro-inflammatory relays between them, are reminiscent of auto-immune chronic inflammatory diseases such as rheumatoid arthritis and Crohn’s disease [5]. These relays lead to the production, inter alia, of pro-inflammatory and inflammatory cytokines such as tumor necrosis factor (TNF), interleukin (IL)-1β, IL-22, and IL-17. Moreover, some of these inflammatory mediators stimulate the proliferation of keratinocytes, which in turn sustain the immune dysregulation of the skin [6]. This general inflammatory background in the skin is responsible for a thickening of both the epidermis (acanthosis) and the stratum corneum (SC, hyperkeratosis). The accelerated renewal of keratinocytes at the SC level is accompanied by a lack of their terminal differentiation, as evidenced by the persistence of their nucleus in the upper layers of the SC (parakeratosis) [2]. As a consequence, the barrier function of the skin is strongly impaired, leading to itch, pain, and bleeding at the psoriatic plaques. Treatment of psoriasis relies on a graduated strategy to resolve inflammation during the crises [3]. The local administration of non-steroidal anti-inflammatory drugs (NSAID) or steroidal cortisone-like drugs is one of the most common prescription for mild to moderate psoriasis. Moderate to severe forms may require phototherapy and/or systemic immuno-suppressive drugs. The latter are mainly engineered injectable biologics that selectively target immune cells or signaling mediators (typically monoclonal antibodies or soluble receptors). Although they have proved their efficacy, some patients do not respond to these therapies, while others develop adverse effects leading to discontinuation across biologic treatments [7,8]. They are also often criticized due to their cost. Therefore, there is an unmet clinical need for the treatment of psoriasis, and innovative therapeutic approaches have to be proposed [3].

Animal models are of great help both to decipher the pathophysiological mechanisms of diseases and to assess new therapeutic approaches. Several animal models of inflammatory diseases of the skin have been developed, especially in mice [9], with some of them developing psoriasis-like burden [10]. One of the most recent and most commonly used is the imiquimod (IMQ)-induced psoriasis-like mouse model [11]. The repeated topical application of IMQ on the shaved skin of mouse causes skin inflammation, leading to cutaneous manifestations reminiscent of that of human psoriasis. The mechanisms underlying these clinical features are complex, involving the concomitant activation of several immune cells [12], as well as oxidative stress [13]. 

Like for many other biomedical purposes, nanotechnology is thoroughly explored for dermatological applications [14]. In psoriasis, nanoformulations of existing drugs, especially of small organic molecules such as methotrexate or analogues of vitamin D, have been developed and assessed with the aim of enhancing the permeability of skin, and thereby improving the efficacy of these drugs. These nanoformulations with therapeutic cargos are lipid or synthetic polymeric nanostructures [15]. Among the latter, dendrimers are hyperbranched, multivalent, and polyfunctional nanodevices that can optimize targeting of anti-psoriasis drugs to relevant epidermal and dermal sites [16]. Dendrimers are synthesized in a stepwise fashion. Generally, low generation dendrimers have a perfectly defined structure. They are a promising alternative to poorly defined nanoparticles, such as linear polymer or metallic nanoparticles, for biomedical development. Dendrimers are designed from a central core to which one or several series of branched monomers are linked. Each monomer is ended by a branching point that enables the dendritic growth of the molecule by addition of the next generation of branched monomers, leading to a “tree-like” structure. The synthesis ends with the addition of functional groups on the last-added series of branched monomers. The total number of series of branched layers determines the generation of the dendrimer [17]. A few types of dendrimers have shown anti-inflammatory properties per se in chronic and acute inflammatory animal models [17,18]. On our side, we have shown that phosphorus-based dendrimers of the poly(phosphorhydrazone) (PPH) series, capped with azabisphosphonate groups, have strong immuno-modulatory and anti-inflammatory effects. The “lead” candidate in the PPH series is a first-generation dendrimer capped with 12 azabisphosphonate (ABP) groups; therefore, it is called the ABP dendrimer (Figure 1).

We have shown that the ABP dendrimer is able to control chronic and acute inflammatory disorders in animals [19,20,21]. As a common feature, we have shown that these therapeutic effects rely on the activation of IL-10-producing immune cells [22]. The cellular mechanisms underlying these therapeutic effects involves the anti-inflammatory activation of monocytes/macrophages [23] and dendritic cells [24], and the control of the proliferation of pro-inflammatory CD4+ T cells [25] together with the induction of regulatory CD4+ T cells producing IL-10 [23]. Moreover, we have also shown that the same dendrimer is able to promote the proliferation of the anti-tumor Natural Killer (NK) cells [26]. Regarding the structure/activity relationships of PPH dendrimers, we have shown that the chemical composition of the surface [27,28] and the global three-dimensional structure [29,30] of the molecule are key determinants of the bioactivity. 

Here, we challenged the immuno-modulatory and anti-inflammatory properties of the ABP dendrimer in the IMQ-induced mouse model of psoriasis. We show that the topical application of the molecule can control the disease from clinical, histopathological, and immunological points of view.

## 2. Materials and Methods 

### 2.1. Synthesis of the ABP Dendrimer

The ABP dendrimer used in this study was synthesized at the “Laboratoire de Chimie de Coordination” (Toulouse, France), and its synthesis has been already described [31]. The near infra-red (NIR) fluorescent analogue ABP-NIR was afforded by the “Laboratoire de Chimie de Coordination” (A. Oukhrib, A.M. Caminade, C.O. Turrin).

### 2.2. Animals

Studies in animals were conducted in accordance with the principles and procedures outlined by the European convention for the protection of vertebrate animals used for experimentation. The mice were housed in groups of 5 in plastic cages in an air-conditioned room (temperature of 22 ± 2 °C and relative humidity of 50 ± 15%) with 12 h artificial light/12 h dark. Food and drinking water were provided ad libitum.

Male Balb/c mice of 9 to 10 weeks of age were obtained from Janvier (Saint-Berthevin, France) and were acclimatized before starting the experimentation. The in vivo design and procedures were assessed and approved by the local Anexplo Ethical Committee. All animals were managed similarly, with due regard for their well-being, according to prevailing practices. The appearance and behavior of animals were assessed at least daily from the start and until termination of the in vivo study. A clinical score was evaluated every day for each animal. This clinical score was based on three criteria: erythema, scaling, and thickness of the skin. Each criterion was quantified with a 1 to 5 scale: 1 corresponds to “normal”, 2 to “slight”, 3 to “moderate”, 4 to “important”, and 5 to “very important”. It was decided before the study that the following ethical points would be checked daily: hydration and feeding default on a period of 24 h to 48 h, weight loss >20%, hypothermia observed with hypoactive animal, dyspnea with increased respiratory movement, skin wound observed purulent, necrotic or exudative; clinical score at 12 for 3 consecutive days. If one of these endpoints was observed on one animal, the animal would have been euthanized immediately.

For the study, the animals were shaved and depilated (using a commercial depilatory cream) on the back at day -1. Then, the induction of psoriasis was performed on animals by daily administration, from day 0 to day 6 in the morning, of 4 mg of IMQ (equivalent to 80 mg of Aldara 5% cream, Meda AB, Sweden) on the skin of the back of animals. The surface of application was about 4 cm^2^ on the back of the animals, which corresponded to a rectangle of about 1.5 by 2.5 cm. After the deposition of the IMQ-containing cream, it was smoothly massaged with a gloved finger for 5 to 10 seconds. The intravenous (IV) injection of the ABP dendrimer was performed every 3 days (at day 1, 4, and 7) at 10 mg/kg, whereas the topical administration was performed daily, in the afternoon, from day 0 to day 6 by deposition of a volume of 90 µL (equivalent to the two doses of 5 and 50 mg/kg) of the ABP dendrimer in phosphate-buffered saline (PBS) and massage. The ABP dendrimer is highly soluble in aqueous solutions (over 1 g/mL). Therefore, it was completely solubilized in PBS at the highest dose used in this study (deposition volume at a concentration of ABP ≈ 16 mg/mL for the dose at 50 mg/kg, depending on the weight of each mouse). For the study based on IV injections of the ABP dendrimer, we euthanized the mice at day 10, whereas for the study based on topical administration of the ABP dendrimer, we euthanized the mice at day 7.

### 2.3. Skin Penetration Study

The ex vivo study of the penetration of dendrimers on IMQ-induced psoriatic skins was conducted using Franz diffusion cells with a diffusion area of 1.77 cm^2^ (PermeGear, Hellertown, PA, USA). The induction of psoriatic skin by IMQ on the back of mice was achieved as described above. After 5 or 7 days, we sacrificed the animals, and the diseased skin of the back was excised and stored at −80 °C until use. Before the permeation experiment, we thawed the skins to room temperature and then mounted them between the donor and receptor compartments of the Franz diffusion cell with the SC facing the donor compartment. The near infra-red fluorescent analogue (ABP-NIR) of the ABP dendrimer was used for visualizing its penetration in the diseased skin. A solution of the ABP-NIR dendrimer in PBS with a concentration corresponding to the dose of 50 mg/kg was prepared and deposited all over each skin in the donor compartment. A control assay was performed with deposition of PBS on the skin from the back of mice to establish the normal fluorescence of the tissues. The receptor compartment was filled with 12 mL PBS and stirred at 300 rpm. After a 24 h incubation at a temperature of 37 ± 1 °C, we removed the skins and carefully washed them with PBS, and then froze them at −80 °C in an optimal cutting-temperature compound (isopentan). We then cut 10 µm thick cryo-sections from the skin samples using a cryostat (CM1950, Leica, Wetzlar, Germany), and we mounted them using ProLong Gold anti-fade reagent with DAPI (4’,6-diamidino-2-phenylindole; Invitrogen, Thermo Fischer Scientific, Waltham, MA, USA) for nuclear staining. Images of 512 × 512 pixels were acquired with a 63X immersion objective on a Leica SP8 confocal microscope. For the detection of the ABP-NIR dendrimer, the excitation wavelength was 635 nm and the emission wavelengths were between 650 and 800 nm. Images were processed using ImageJ software.

### 2.4. Histopathology

Excised dorsal skins were fixed for 24 h in 4% (*w*/*v*) buffered formalin and embedded in paraffin. Then, 5 µm sections from which paraffin had been removed were stained using routine histological hematoxylin/eosin staining. Images were obtained with a Panoramic Slide Scanner 250 (3DHISTECH, Budapest, Hungary), and all slides were scanned at 40X magnification. Histological analysis and scoring of the slides were performed on the whole section. The histological score was based on the six following parameters: acanthosis, hyperkeratosis, parakeratosis, absence of granular layer, presence of Munro’s micro-abscesses, and infiltration of immune cells. For each, a score ranging from 0 to 1 (0: none; 0.5: moderate; 1: marked) was given and these scores were cumulated to give the histological score of the back.

### 2.5. Immunohistochemistry

After antigen retrieval using either heat-induced epitope retrieval (HIER) in citrate buffer (pH = 6) for T cells and neutrophils, or proteinase K (R.T.U., S3020, Dako) for macrophages, we incubated sections with the following antibodies: the rabbit monoclonal anti-CD3 clone SP7 diluted at 1:200 (ab16669, Abcam) at room temperature (RT) for 1 h, the rat monoclonal anti-polynuclear neutrophil (PNN) marker clone 64608 diluted at 1:200 (sc-71674, Santa-Cruz) at 4 °C overnight, and the rat monoclonal anti-F4/80 clone Cl:A3-1 diluted at 1:100 (MCA497GA, AbD Serotec) at RT for 1 h. Sections were then incubated for 30 min at RT with biotinylated goat-anti-rabbit diluted at 1:250 (R.T.U., BP-9100, Vector Laboratories), or goat-anti-rat diluted at 1:50 (STAR113P, Bio-Rad) secondary antibodies followed by incubation with horseradish peroxidase (HRP) (R.T.U., PK-7100, Vector Laboratories) for 30 min. Diaminobenzidine (SK-4105, Vector Laboratories) was used to reveal the staining, followed by blue hematoxylin counterstaining. 

Quantification of immunohistochemistry staining was done in 5 zones of similar areas in all sections of dorsal skin. HistoQuant software (3DHISTECH) was used to quantify the number of CD3+ T lymphocytes and PNN per area (cells/mm^2^), and DensitoQuant software (3DHISTECH) was applied to calculate the percentage of all F4/80-positive pixels (sum of weak, moderate, and strong positive pixels divided by the total number of pixels).

### 2.6. Statistical Analyses

All statistical analyses were performed with Prism 5.0 (GraphPad, San Diego, CA, USA). The homogeneity of the variances of the 3 groups was analyzed using the Bartlett’s parametric test. For Figure 4C,D and Figure 6A–C, we verified the homogeneity of the variances. Therefore, the one-way ANOVA test was used followed by the post hoc Dunnett’s test to evidence statistically significant differences. On the contrary, for Figure 5D,E, we found the variances to not be homogeneous. Therefore, we applied the non-parametric Kruskal–Wallis’ test followed by the post hoc Dunn’s test to evidence statistically significant differences.

## 3. Results

### 3.1. Clinical Assessment of the Efficacy of the ABP Dendrimer

In a first assay using the mouse model in which psoriasis-like disease is induced by daily application of IMQ on the dorsal skin of mice, we administered the ABP dendrimer via the systemic IV route of administration. The ABP dendrimer was injected at 10 mg/kg at days 1, 4, and 7 (Figure 2A). The clinical score was assessed daily from day 0 until day 10, and no difference was observed between the two groups of mice during the assay. Results obtained at day 10 are shown in Figure 2B.

Then, in a second assay, we moved to topical administration of the ABP dendrimer. Prior to the study, we assessed the capability of the ABP dendrimer to permeate the skin of IMQ-induced psoriatic mice. We took advantage of an already available fluorescent analogue of the ABP dendrimer that emits near infra-red fluorescence thanks to the presence of a polymetine-based mega-Stokes dye in place of one of the six branches of the ABP dendrimer (the so-called ABP-NIR dendrimer) [32]. The near infra-red fluorescence does not overlap auto-fluorescence of the tissues. Diseased skins were excised from the back of mice at days 5 and 7. Using a Franz diffusion cell, we tested the penetration of the ABP-NIR dendrimer at the highest dose intended for the topical application of the ABP dendrimer (50 mg/kg). To better appreciate the penetration of the ABP-NIR dendrimer in the skin, we imaged co-staining of the ABP-NIR dendrimer and DAPI, a nuclear fluorescent probe. Figure 3A shows the normal fluorescence of the skin tissues at the wavelengths used for imaging. After both 5 days (Figure 3B, two mice) and 7 days (Figure 3C, two mice), the ABP-NIR dendrimer penetrated the psoriatic skin beyond the SC with a clear co-localization with the nucleated cells of epidermis and dermis.

For the assay with topical application, we chose two doses of the ABP dendrimer, which were administered daily via massage of the skin of the back (5 and 50 mg/kg in PBS) (Figure 4A). Figure 4B presents representative pictures of the dorsal skin of mice. These pictures reveal a perceivable visual improvement of the skin of treated mice when compared to control psoriatic mice, especially in the group treated at the lower dose. The skin looked thinner and there were significantly less scales. Indeed, the clinical score at day 6 was significantly lower with either doses of the ABP dendrimer, with the difference being more significant at 5 mg/kg (Figure 4C). The cumulative scores from day 0 to day 7 also showed the same statistically significant differences between the control psoriatic group and the treated groups (Figure 4D).

### 3.2. Histological Assessment of the Efficacy of the ABP Dendrimer

At day 7, mice were sacrificed and the dorsal skin was excised for histological analysis after hematoxylin/eosin staining. Figure 5A shows the classical histopathological features of a psoriatic skin. At the SC level, hyperkeratosis (thickening of the SC), parakeratosis (retention of nuclei in the upper layers of keratinocytes of the SC), and a diminished granular layer at the basis of SC were noticeable. However, Munro’s micro-abscesses were scarce. At the epidermal level, acanthosis (diffuse epidermal hyperplasia), and spongiosis (intercellular edema) were observed, with both being responsible for a thickening of the epidermis. Finally, at the dermal level, a dense infiltration of immune cells was evidenced.

The same analysis performed on the dorsal skin of mice treated with the ABP dendrimer underscored a lower hyperkeratosis and no parakeratosis (this combination being defined as orthokeratosis) at the SC level. Moreover, the granular layer was perfectly visible. At the epidermal level, acanthosis and spongiosis were decreased, whereas at the dermis level, the infiltration of immune cells was weaker (Figure 5B,C). These histopathological features were quantified to establish a histological score for each mouse. Results are presented in Figure 5D and show a statistically significant decrease of the histological score only in the group of mice treated with 50 mg/kg of ABP dendrimer. On the contrary, measurement of the thickness of the skin of the back showed a significant decrease only in the group of mice treated with 5 mg/kg of ABP dendrimer, as reported for day 6 (Figure 5E).

### 3.3. Immunological Assessment of the Efficacy of the ABP Dendrimer

The skin of the back of mice were also used for the quantification of the infiltration of immune cells by immunohistochemistry. Three immune populations were quantified: F4/80-positive cells (mainly infiltrating macrophages, but also dendritic cells and resident Langerhans’ cells [33]), CD3-positive cells (i.e., T lymphocytes), and polynuclear neutrophils (PNN). Considering that the F4/80 staining was very dense and that F4/80-positive cells are spread out and flattened, we decided to quantify them as the ratio of the number of F4/80-positive pixels to the total number of pixels on the slide (Figure 6A).

For the two other cell populations (T cells and PNN), we were able to perform cell counting, which we reported as the number of cells per square millimeter (Figure 6B,C). Only F4/80-positive cells were statistically decreased in both treated groups of mice, with the decrease being more significant at 5 mg/kg of ABP dendrimer (Figure 6A). Moreover, a slight decrease in the number of T lymphocytes, although not statistically significant, was noticed (Figure 6B). Below each graph of Figure 6, representative pictures of the immunohistochemistry slides from control psoriatic mice, as well as mice treated with both 5 and 50 mg/kg of the ABP dendrimer, are shown.

## 4. Discussion

For many years, dendrimers have been the subject of applicative research in the biomedical field, mainly as agents for imaging and nucleic acid transfection, and also as drug delivery vectors. More recently, some dendrimers have emerged as efficient drug candidates by themselves, especially for the treatment of inflammatory and infectious contexts [17,34]. In particular, the ABP dendrimer used in this study showed dramatic therapeutic effects in two mouse models of experimental arthritis relevant to rheumatoid arthritis (RA) [19,30], and in a mouse model of experimental autoimmune encephalomyelitis (EAE) relevant to multiple sclerosis (MS) [20]. As these two diseases are chronic inflammatory diseases of autoimmune origin, we assessed the therapeutic effect of the ABP dendrimer in a mouse model of psoriasis. However, contrarily to RA and MS, the autoimmune origin of psoriasis remains a key question, and it is now acknowledged that this illness is not a bona fide autoimmune disease [2,4]. In a first attempt with the IMQ-induced psoriatic mouse model, we administered the ABP dendrimer intravenously, as we did before for RA and MS mouse models, at the dose of 10 mg/kg every 3 days. This protocol did not afford any improvement of the disease status. This absence of any beneficial effect was unexpected as the dermis is vascularized, with blood capillaries vascularizing the papillary dermis. This uppermost layer of the dermis is intertwined with the rete ridges of the epidermis that project into the papillary dermis. Moreover, during psoriasis, there is both a distention of existing blood vessels and neovascularization at the papillary dermis level [3]. These pathophysiological features should have favored the IV route of administration. Therefore, someone’s first intention could have been to repeat the IV protocol with increasing doses of the ABP dendrimer combined with a higher frequency of administration. However, we assayed the topical route of administration through daily massages with the ABP dendrimer in PBS. We tested two doses of the molecule corresponding to 5 and 50 mg/kg, starting from the initial principle that the active ingredient is totally absorbed through the skin barrier, which was probably not the case. Nevertheless, via the topical route of administration and with either doses, we showed that the ABP dendrimer had a statistically significant therapeutic effect in the IMQ-induced psoriatic mouse model. However, one should notice that, first, there was no advantage of the highest dose of 50 mg/kg, and secondly the efficacy of the ABP dendrimer in this model was not as dramatic as its efficacy shown previously in RA and MS mouse models in which the clinical score was reversed almost back to zero [19,20,30]. This can be explained by the difficulty for the ABP dendrimer to cross the skin barrier in sufficient amounts, even if the latter is known to be impaired in psoriatic skin. Nevertheless, consistent with the clinical score, the histopathological and the immune infiltration analyses confirmed the therapeutic benefit of the ABP dendrimer at both doses of 5 and 50 mg/kg. Indeed, all histopathological features of the disease (hyperkeratosis, parakeratosis, the absence of the granular layer at the SC level, acanthosis, spongiosis, and infiltration of immune cells in the dermis) were improved upon treatment with the ABP dendrimer. There again, there was no advantage of the highest dose.

To our knowledge, this is the first time that the therapeutic effect of a dendrimer per se is shown in a mouse model of psoriasis. However, some dendrimers have been used to enhance the delivery of drugs through the skin barrier. As recently as 2003, Chauhan et al. showed that poly(amidoamine) (PAMAM) dendrimers of the fourth generation (G4) that ended with either –NH_2_ or –OH functions enhanced the skin permeation of the model hydrophobic drug indomethacin [35]. The dendrimers enable this enhancement as they carry the drug molecules to the surface of the skin in a solubilized form from which indomethacin will partition to the hydrophobic environment of the SC. It is likely that due to their molecular weight (MW, between 14 and 15 kDa) and hydrophilicity, the G4 PAMAM dendrimers used in this study cannot permeate the skin. Later on, it has been shown that G3 and G4 PAMAM dendrimers also enhance skin permeation of 8-methoxypsoralene, an anti-psoriatic drug used in phototherapy of the disease [36]. More recently, a polypropylene imine (PPI) dendrimer of the fifth generation harboring –NH_2_ functions at its surface was used to deliver the anti-psoriatic drug dithranol more efficiently, and also to protect the drug from light-induced degradation and the skin from irritation, burning sensation, and staining [16]. This delivery system based on a PPI dendrimer was improved by incorporating it in a microsponge-based gel formulation [37]. This strategy was implemented to overcome the troublesome and inconvenient application of dithranol.

The therapeutic efficacy of an anti-psoriatic drug delivered topically is conditioned by its skin permeation capability. Even if the skin barrier function of psoriatic skin is impaired, we acknowledge that only small hydrophobic molecules can efficiently reach the epidermis and dermis. Nevertheless, the heterogeneous structures of the SC, as well as the delivery vehicle itself, are important to predict the skin permeation of a multicomponent system [38]. Interestingly, it has been shown that diverse PAMAM dendrimers varying in size (i.e., generation), surface function (neutral vs. anionic), and concentration can enhance the permeability of skin when they are applied to the skin before topical administration of a model hydrophilic drug such as 5-fluorouracyl [39]. The same property of a low generation G2 PAMAM dendrimer has been proven, as its application to the skin before the topical administration of salicylic acid enhances the partition of the latter to SC [40]. Salicylic acid has been used for a long time because of its keratolytic effect enabling further penetration of anti-psoriatic drugs [41]. More recently, the skin penetration properties of PAMAM dendrimers have been rationalized [42], concluding that the smallest G2 cationic amine-capped dendrimer (MW = 3256 Da) show deeper penetration of the skin. PAMAM dendrimers are fully hydrophilic compounds. On the contrary, we have shown that PPH dendrimers such as the ABP dendrimer adopt a directional conformation in physiological aqueous medium, unexpectedly exposing their hydrophobic core and branched monomers to the environment [29,30]. This three-dimensional particularity, which should be adequate for the ABP dendrimer to exert its anti-inflammatory effects in the psoriatic skin, is probably counterbalanced by the anionic ABP groups gathered on the other side of the molecule, explaining the moderate therapeutic effect of the ABP dendrimer in the IMQ-induced mouse model of psoriasis.

## 5. Conclusions

Herein, we show for the first time that a dendrimer has anti-psoriatic therapeutic effects by itself in a mouse model of the disease. The anti-inflammatory ABP dendrimer was found to have therapeutic effects in animal models of acute and chronic inflammatory disorders when it was administered either systematically or loco-regionally. Moreover, its preclinical tolerability and safety profiles in various animal models [32,43] make it a promising candidate for chronic inflammatory diseases, including psoriasis. However, the therapeutic efficacy of the ABP dendrimer to treat psoriatic mouse model has to be increased. For this purpose, smart pre-formulation systems have to be designed and evaluated to enhance the penetration of the drug to the epidermis and dermis of the diseased skin.

## Figures and Tables

**Figure 1 biomolecules-10-00949-f001:**
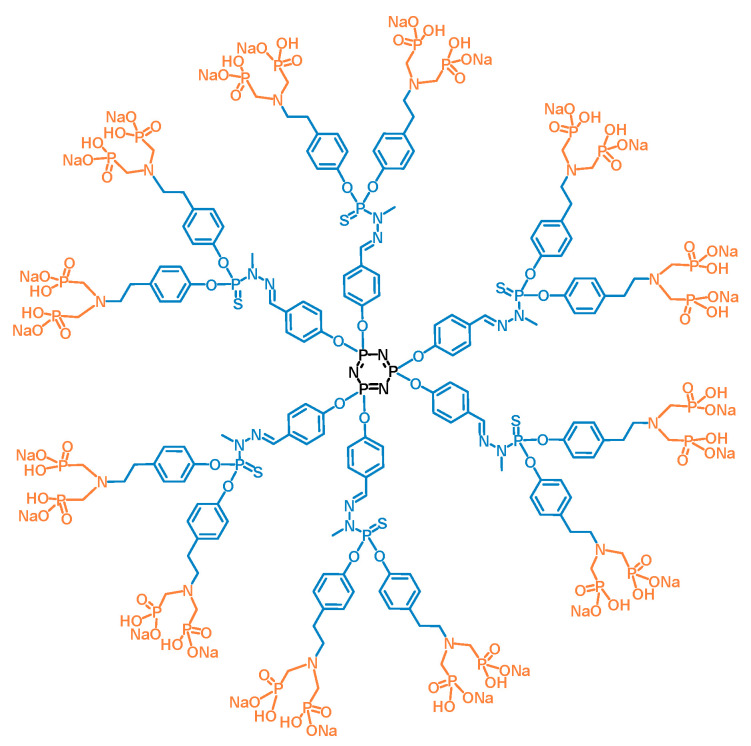
Two-dimensional structure of the azabisphosphonate (ABP) dendrimer. The cyclotriphosphazene core (N_3_P_3_) is in black, and the poly(phosphorhydrazone) (PPH) branched monomers (including the point of divergence) are in blue. The tyramine-based (in blue) ABP surface groups are in orange.

**Figure 2 biomolecules-10-00949-f002:**
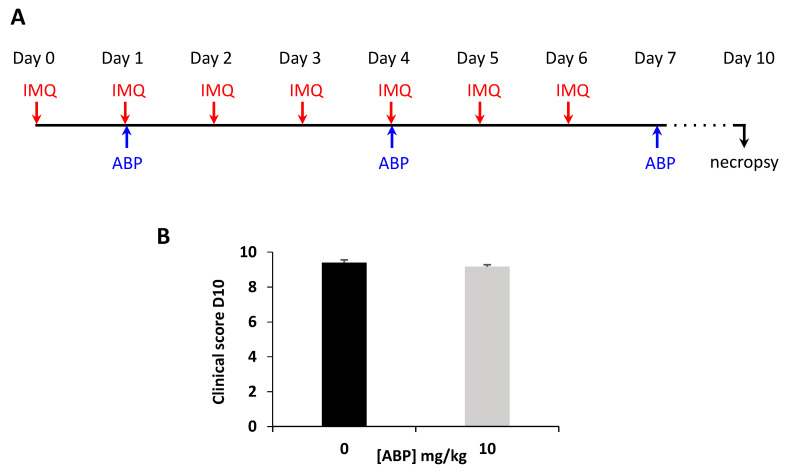
Clinical assessment of the efficacy of the ABP dendrimer when administered intravenously to imiquimod (IMQ)-induced psoriatic mice. (**A**) Time line of the study. (**B**) Clinical score at day 10 for control psoriatic mice (black bar) and mice treated with 10 mg/kg of the ABP dendrimer every 3 days (*n* = 10 mice in each group).

**Figure 3 biomolecules-10-00949-f003:**
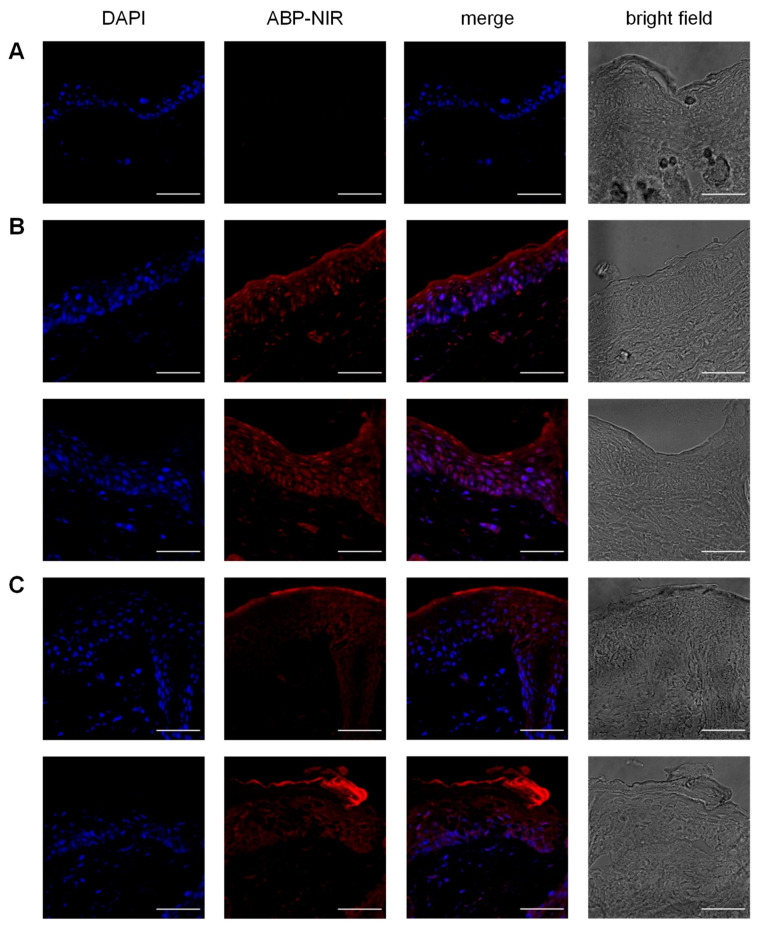
Penetration of the IMQ-induced psoriatic skin by the near infra-red (NIR) ABP-NIR dendrimer, a fluorescent analogue of the ABP dendrimer. First column shows the nuclear 4’,6-diamidino-2-phenylindole (DAPI) staining, second column shows the near infra-red fluorescence of the ABP-NIR dendrimer, third column shows the merge of the first two columns, and last column shows bright field imaging of (**A**) skin incubated with phosphate-buffered saline (PBS; control), and (**B**) day 5 and (**C**) day 7 psoriatic skins incubated with the ABP-NIR dendrimer (two mice each). Scale bars represent 50 µm.

**Figure 4 biomolecules-10-00949-f004:**
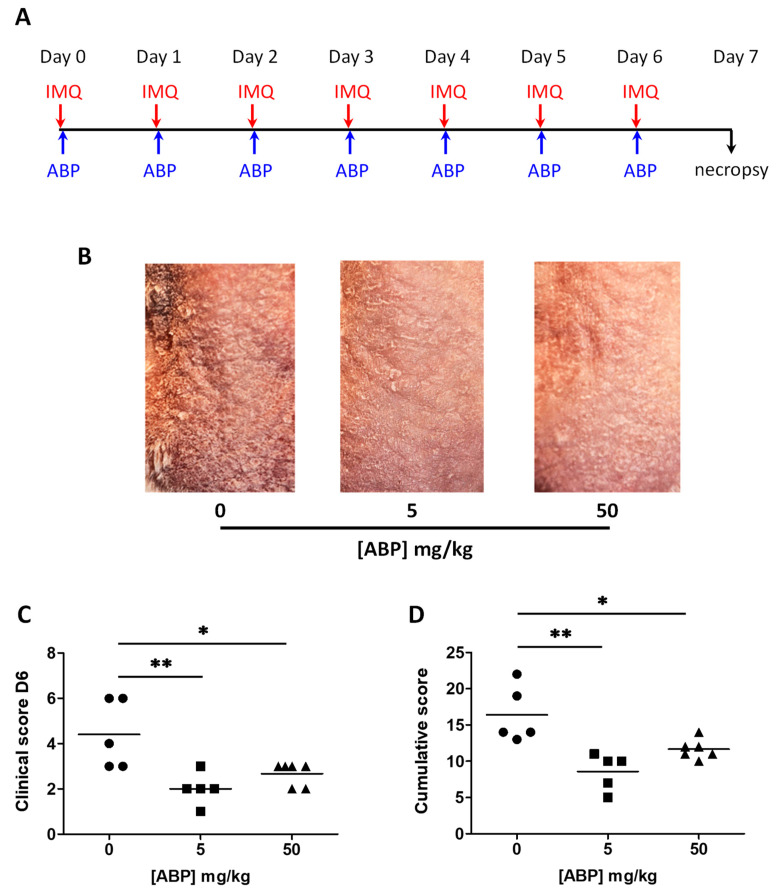
Clinical assessment of the efficacy of the ABP dendrimer when administered topically to IMQ-induced psoriatic mice. (**A**) Time line of the study. (**B**) Representative pictures of the skin of the back of mice. On the left, control psoriatic mouse; in the middle and on the right, mouse treated daily with 5 mg/kg and 50 mg/kg of the ABP dendrimer, respectively. (**C**) Clinical score at day 6 for each group of mice (*n* = 5 mice in the control group at a concentration of ABP = 0 mg/kg, *n* = 5 mice for the group at 5 mg/kg, and *n* = 6 mice for the group at 50 mg/kg). (**D**) Cumulative clinical score from day 1 to day 7 for each group of mice. * *p* < 0.05, and ** *p* < 0.01 versus untreated mice (concentration of ABP = 0 mg/kg) using the one-way ANOVA test followed by the post hoc Dunnett’s test.

**Figure 5 biomolecules-10-00949-f005:**
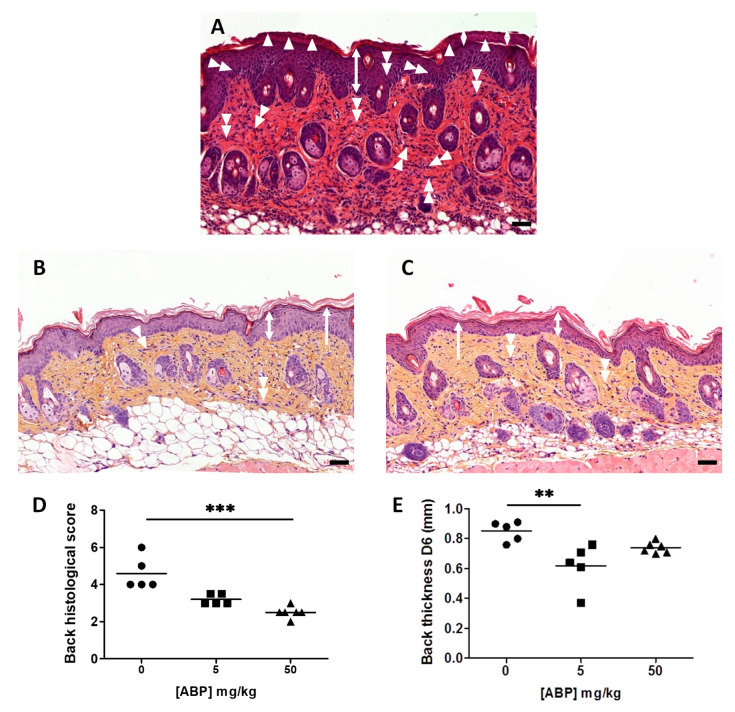
Histological assessment of the efficacy of the ABP dendrimer when administered topically to IMQ-induced psoriatic mice. Representative pictures of the skin of the back of mice stained with hematoxylin/eosin at day 7 from psoriatic control mouse (representative of *n* = 5 mice) (**A**), and from mice treated daily with 5 mg/kg (representative of *n* = 5 mice) (**B**) and 50 mg/kg (representative of *n* = 6 mice) (**C**) of the ABP dendrimer; black scale bars represent 50 µm. Single-headed arrows point the granular layer (absent in (**A**)), double-headed arrows show acanthosis, single arrow heads show parakeratosis (lowered in (**B**) and (**C**)), diamonds show hyperkeratosis (lowered in (**B**) and (**C**)), double arrow heads show infiltration of immune cells. (**D**) Scoring of the histological assessment of the skin of the back for each group of mice; *** *p* < 0.001 versus untreated mice (concentration of ABP = 0 mg/kg) using the Kruskal–Wallis’ test followed by the post hoc Dunn’s test. (**E**) Measurement of the thickness of the skin of the back for each group of mice at day 6; ** *p* < 0.01 versus untreated mice (concentration of ABP = 0 mg/kg) using the Kruskal–Wallis’ test followed by the post hoc Dunn’s test.

**Figure 6 biomolecules-10-00949-f006:**
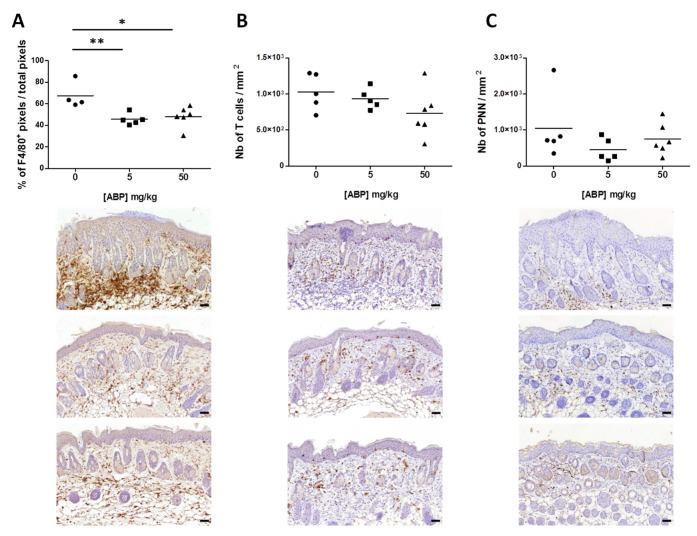
Immunological assessment of the efficacy of the ABP dendrimer when administered topically to IMQ-induced psoriatic mice. Quantification of the infiltration of F4/80-positive cells (mainly macrophages) (**A**), T lymphocytes (**B**), and polynuclear neutrophils (**C**) in the skin of the back of each group of mice (*n* = 5 mice in the control group at a concentration of ABP = 0 mg/kg, *n* = 5 mice for the group at 5 mg/kg, and *n* = 6 mice for the group at 50 mg/kg) by immunohistochemistry; * *p* < 0.05, and ** *p* < 0.01 versus untreated mice (concentration of ABP = 0 mg/kg) using the one-way ANOVA test followed by the post hoc Dunnett’s test. Below each graph, representative immunochemistry pictures of the skin of the back of mice at day 7 from psoriatic control mouse, and mice treated daily with 5 mg/kg and 50 mg/kg of the ABP dendrimer, from top to bottom, respectively, are shown; black scale bars represent 50 µm.

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
