# Peer review of "An Anti-Inflammatory Poly(PhosphorHydrazone) Dendrimer Capped with AzaBisPhosphonate Groups to Treat Psoriasis"

_biomolecules, 2020, doi:10.3390/biom10060949_

Round 1

Reviewer 1 Report

In the present manuscript, authors study the possible effect of Poly(PhosphorHydrazone) dendrimer capped with AzaBisPhosphonate groups in a model of psoriasis in mice.

Results are quite interesting however the following points must be clarified:

First of all, authors report that topical application of ABP dendrimer solution in PBS may have some effects in symptom control, without difference between two doses. However, likely the compound is not absorbed through the skin, as they also discuss (lines 301 – 304) but they report a light beneficial effect. However, I found some incongruence indeed, while they discuss that the effect is very light ( lines 306, line 353) and in some way non-specific, in the title, in the abstract and introduction this is not clear, they state (lines 115 -117) “……show that the topical application of the molecule is efficient to control the disease from clinical, histopathological, and immunological points of view”.

Furthermore, in my opinion the effect observed might be due to a barrier effect of topical application. Was the compound tested completely solubilized in PBS? I hope it was, because if this is not the case, a barrier effect is very probable.

It would be valuable to evaluate skin penetration of compound

Control should be performed with not capped PPH, or with an inert dendrimer.

The n value (number of cases) for each experiment is missed and must be reported

A reference drug applied topically, for example beclomethasone, could be used to which compare the effect of beneficial effects ABP dendrimer solution

Author Response

REVIEWER 1

In the present manuscript, authors study the possible effect of Poly(PhosphorHydrazone) dendrimer capped with AzaBisPhosphonate groups in a model of psoriasis in mice.

Results are quite interesting however the following points must be clarified:

First of all, authors report that topical application of ABP dendrimer solution in PBS may have some effects in symptom control, without difference between two doses. However, likely the compound is not absorbed through the skin, as they also discuss (lines 301 – 304) but they report a light beneficial effect. However, I found some incongruence indeed, while they discuss that the effect is very light ( lines 306, line 353) and in some way non-specific, in the title, in the abstract and introduction this is not clear, they state (lines 115 -117) “……show that the topical application of the molecule is efficient to control the disease from clinical, histopathological, and immunological points of view”.

We thank the reviewer for sharing her / his feeling. We have submitted a pioneering study as it is the first time that a dendrimer shows anti-inflammatory effects by itself on inflammatory skin disease through topical application. Regarding the clinical, histopathological, and immunological features of the diseases, we have shown that the effects of the molecule are statistically significant. Therefore, we do believe that the ABP dendrimer has beneficial therapeutic effects in the IMQ-induced psoriatic mouse model. However, we agree that the effects of the molecule are not dramatic, and we do hope that we will be able to improve them with optimized analogues of the ABP dendrimer and smart formulations. Consistently, we propose to remove the word “efficient” from the lines 155-117. Accordingly, we have re-written the last sentence of the introduction.

Furthermore, in my opinion the effect observed might be due to a barrier effect of topical application. Was the compound tested completely solubilized in PBS? I hope it was, because if this is not the case, a barrier effect is very probable.

This is indeed a very smart point we forgot to mention in the manuscript. The ABP dendrimer is highly soluble in aqueous solutions (over 1 g/mL). Therefore, it was completely solubilized in PBS at the highest dose used in this study (deposition volume at [ABP] = 16 mg/mL for the dose at 50 mg/kg, depending on the weight of each mouse). We have added this statement in paragraph 2.2. of the Materials and Methods section.

It would be valuable to evaluate skin penetration of compound

Thank you very much for this proposal. We have taken advantage of a fluorescent analogue of the ABP dendrimer to show that such molecules can penetrate the psoriatic mouse skin ex vivo (using Franz’ diffusion cell). Results are shown in the new Figure 3, and described and commented in the paragraph 3.1 of the Results section. We hope that the reviewer will be convinced by the data.

Control should be performed with not capped PPH, or with an inert dendrimer.

We thank the reviewer for this remark. In the submitted study we have assessed for the first time the topical anti-inflammatory effect of a dendrimer. The objective is not yet to delineate the structural requirements for this new bio-activity of the dendrimer. Although relevant, this point should be address in a next series of experiments to unveil the cellular and molecular effects of the dendrimer on IMQ-induced psoriatic mice.

The n value (number of cases) for each experiment is missed and must be reported

We have reported the number of mice for each experiment in the legend of Figures 2, 3, 4, 5, and 6 of the revised version of the manuscript.

A reference drug applied topically, for example beclomethasone, could be used to which compare the effect of beneficial effects ABP dendrimer solution

Thank you very much for this sounded concern. We have chosen not to include such a control for two reasons:

  • The reference drugs (such as beclomethasone or dexamethasone) that are commercially available for topical application are formulated in creams. These formulations are optimized both for skin penetration and enhancement of the therapeutic efficacy of the drug. The aim of the submitted study is to prove the therapeutic efficacy of the ABP dendrimer by itself, therefore it would not have been fair to challenge the effect of the tested item with a formulated standard.
  • We have benchmarked recent publications in the field. Over the 16 most recent articles showing topical efficacy of new compounds, 5 have not included such a control (published in Sci. Rep., Int. J. Pharm., Int. Immunopharmacol., Antioxid. Redox Signal., Arch. Biochem. Biophys.).

Reviewer 2 Report

The manuscript describes investigations of immuno-modulatory and anti-inflammatory properties of the ABP dendrimer after topical application in the IMQ-induced mouse model of psoriasis. Several concerns needs to be addressed. My specific comments are as follows:

  1. Line 20-21: Abstract first sentence “~ which size and structure are perfectly controlled.” is not appropriate because dendrimer chemistry can be often not perfect and is uncontrolled. Several studies point out the defect in the structure of dendrimer.
  2. Line 26~28: The content does not match text because oral and vitreous tissue applications are not found in the experiments and results.
  3. Line 85, As commented above, “perfectly” is not appropriate.
  4. Did you formulate ABP dendrimers for topical application? According to the methods, it is shown that PBS solution containing ABP dendrimers was applied to the skin. If the ABP dendrimer was formulated for transdermal delivery, it would have produced different results. Have you tried any formulation approaches?
  5. This study needs to confirm that ABP dendrimers can penetrate the skin. Have you ever tried to identify the absorption of ABP dendrimers through the skin?
  6. Line 356: The authors stated that ABP dendrimer has anti-psoriatic therapeutic effects by itself in a mouse model of the disease. But, the reviewer feels that this study is not sufficient to state anti-psoriatic therapeutic effects of ABP dendrimer.

Author Response

REVIEWER 2

The manuscript describes investigations of immuno-modulatory and anti-inflammatory properties of the ABP dendrimer after topical application in the IMQ-induced mouse model of psoriasis. Several concerns needs to be addressed. My specific comments are as follows:

1. Line 20-21: Abstract first sentence “~ which size and structure are perfectly controlled.” is not appropriate because dendrimer chemistry can be often not perfect and is uncontrolled. Several studies point out the defect in the structure of dendrimer.

We would like to highlight that the ABP dendrimer is a small, first generation phosphorus-based dendrimer. Several years ago, we published the synthesis and the characterization of this molecule. The stepwise synthesis of this compound affords isomolecular batches, and the structure of the molecule is easily controlled using 31P NMR. Nevertheless, as it is true that for high generation dendrimers, defects in the structures occur, we have modulated the first sentence of the abstract.

2. Line 26~28: The content does not match text because oral and vitreous tissue applications are not found in the experiments and results.

This part of the abstract refers to the background of the submitted study. We think that it is important to explain to the readers that in previous studies we have already assessed different routes of administrations in animal models of inflammatory disorders. Here for the first time we have tested the topical route of administration of an anti-inflammatory dendrimer to treat psoriatic mice.

3. Line 85, As commented above, “perfectly” is not appropriate.

In agreement with both the reviewer’s concern and the points mentioned in our answer to the reviewer’s first point, we have restricted the statement to small, low generation dendrimers.

4. Did you formulate ABP dendrimers for topical application? According to the methods, it is shown that PBS solution containing ABP dendrimers was applied to the skin. If the ABP dendrimer was formulated for transdermal delivery, it would have produced different results. Have you tried any formulation approaches?

Thank you very much for these questions. As stated in the Materials and Methods section, the ABP dendrimer is simply solubilized in PBS. Indeed, the aim of the submitted study is to prove the therapeutic efficacy of the ABP dendrimer by itself, without any formulation to optimize its skin penetration or to enhance its therapeutic efficacy. The formulation of such a compound (which is not hydrophobic as the vast majority of drugs) is very challenging. We have mentioned in the “5. Conclusions” section that further perspectives can include the formulation of the ABP dendrimer to improve its efficacy on psoriasis.

5. This study needs to confirm that ABP dendrimers can penetrate the skin. Have you ever tried to identify the absorption of ABP dendrimers through the skin?

Thank you very much for this proposal. We have taken advantage of a fluorescent analogue of the ABP dendrimer to show that such molecules can penetrate the psoriatic mouse skin ex vivo (using Franz’ diffusion cell). Results are shown in the new Figure 3, and described and commented in the paragraph 3.1 of the Results section. We hope that the reviewer will be convinced by the data.

6. Line 356: The authors stated that ABP dendrimer has anti-psoriatic therapeutic effects by itself in a mouse model of the disease. But, the reviewer feels that this study is not sufficient to state anti-psoriatic therapeutic effects of ABP dendrimer.

We state that the ABP dendrimer has anti-psoriatic therapeutic effects by itself because it dampens both clinical, histopathological, and immunological features of the disease. Moreover, as suggested, we have added data showing that this type of molecule is absorbed through the skin, which is consistent with the therapeutic benefit we have shown.

Round 2

Reviewer 1 Report

The manuscript has been improved. Authors have replied satisfactorily to all my queries.